# Novelty Search Promotes Antigenic Diversity in Microbial Pathogens

**DOI:** 10.3390/pathogens12030388

**Published:** 2023-02-28

**Authors:** Brandon Ely, Winston Koh, Eamen Ho, Tasmina M. Hassan, Anh V. Pham, Weigang Qiu

**Affiliations:** 1Department of Biology, Graduate Center, City University of New York, New York, NY 10016, USA; 2Department of Biological Sciences, Hunter College, City University of New York, New York, NY 10065, USA; 3Department of Physiology and Biophysics, Institute for Computational Biomedicine, Weill Cornell Medical College, New York, NY 10021, USA

**Keywords:** surface antigen, antigenic variation, fitness landscape, evolution, genetic algorithm, novelty search, NK model, local optima networks

## Abstract

Driven by host–pathogen coevolution, cell surface antigens are often the fastest evolving parts of a microbial pathogen. The persistent evolutionary impetus for novel antigen variants suggests the utility of novelty-seeking algorithms in predicting antigen diversification in microbial pathogens. In contrast to traditional genetic algorithms maximizing variant fitness, novelty-seeking algorithms optimize variant novelty. Here, we designed and implemented three evolutionary algorithms (fitness-seeking, novelty-seeking, and hybrid) and evaluated their performances in 10 simulated and 2 empirically derived antigen fitness landscapes. The hybrid walks combining fitness- and novelty-seeking strategies overcame the limitations of each algorithm alone, and consistently reached global fitness peaks. Thus, hybrid walks provide a model for microbial pathogens escaping host immunity without compromising variant fitness. Biological processes facilitating novelty-seeking evolution in natural pathogen populations include hypermutability, recombination, wide dispersal, and immune-compromised hosts. The high efficiency of the hybrid algorithm improves the evolutionary predictability of novel antigen variants. We propose the design of escape-proof vaccines based on high-fitness variants covering a majority of the basins of attraction on the fitness landscape representing all potential variants of a microbial antigen.

## 1. Introduction

### 1.1. Antigenic Diversification

The emergence of novel variants of a single pathogen species poses a challenge in controlling pathogen transmission within a host population and in the development of vaccines and diagnostic tests [1,2,3]. The strain theory posits that despite the existence of multiple mechanisms for genetic exchange by the pathogen within the host, distinct strains are maintained within the population, with a strain being defined by the specific set of loci with the greatest influence on transmission [4]. Often, these loci are antigens. In an evolutionary dynamic known as the “Red Queen” hypothesis, antigenic diversification is driven by strong selective pressure from the host immune system [5,6]. Thus, while microbial pathogens may appear to be limitless in varying their antigenic profiles, evolutionary diversification of microbial antigens is constrained by at least two biological tradeoffs. First, all antigen variants need to be structurally and functionally intact, if not enhanced, e.g., in binding a new host receptor. Second, antigen variants need to be immunologically distinct to escape herd immunity in the host population. One manifestation of these tradeoffs is that antigenic diversification often occurs at multiple loci within the same species, and antigen variants often differ greatly in molecular sequences [7,8,9]. Various models of immune selection have theorized causes for this phenomenon, such as enhanced immune evasion when different strains are equipped with non-overlapping sets of antigenic alleles, heterogeneity in antigen targeting by different hosts, and selection for novel antigen function [4,7].

### 1.2. Fitness Landscapes

Antigen diversification in a microbial pathogen can be conceptualized, modeled, and predicted with fitness landscapes. The idea of a fitness (or adaptive) landscape was first used as a metaphor by Sewall Wright in 1932 to describe the relationship between genotype and fitness [10]. Akin to topographic maps, these three-dimensional fitness landscapes plot possible genotype combinations along the *x* and *y* axes, and fitness (or a fitness proxy) on the *z* axis. The genotypic space on the map can have varying degrees of ruggedness, with multiple peaks and valleys representing the highest and lowest local fitness levels of given genotypes, respectively. Navigation through the space is representative of evolution, while climbing peaks is analogous to adaptation. In a fitness landscape of a microbial antigen, the genotypic axes are represented by sequences (or haplotypes, if only epitope sites are considered) of all potential antigen variants, and the fitness axis is represented by a functional metric (e.g., receptor affinity) of the variants. In an antigen fitness landscape, the genotypic space is hyperdimensional, with each axis representing variability at a single haplotype position. The ability to construct and analyze fitness landscapes paves the way toward evolutionary predictability [11].

Long after the conception of Wright’s fitness landscapes, Stuart Kauffman and colleagues proposed a mathematical model for analyzing the structure of these landscapes, the NK model [12]. The model has two main parameters: *N*, the number of components in a system, and *K*, the degree of interaction between components. Although this model has various applications, its original inception was rooted in biology. In the case of protein space, *N* is equivalent to the number of amino acids in the protein sequence, while *K* represents the degree of epistatic interactions between the amino acids. Applying this model to the maturation of an immune response, Kauffman showed how different parameters of the model can be set to simulate observed conditions during the affinity maturation of antibodies. To tune the ruggedness of the landscape, *K* was set so that the average walk length from a likely starting position to a local optimum was equivalent to the average walk lengths observed in actual affinity landscapes. This model can yield several insights for the fitness landscape, including ruggedness and the distance and accessibility to fitness peaks. NK landscapes are considered an acceptable model for rugged landscapes [13,14] and have since been used to explore landscape structure and test evolutionary algorithms [15,16,17,18].

Until recently, models of fitness landscapes were confined to theoretical analyses. With the development of methods like deep mutational scanning (DMS), the robust datasets required to construct empirical fitness landscapes are now attainable [19,20]. Deep mutational scans can facilitate assays to link genotypes to phenotypes for thousands of protein variants at one time, revealing information on the effects of specific mutations, protein structure, and epistatic interactions between amino acids in a peptide sequence. Since its inception, several investigations have utilized DMS to study the effects of mutations on antigen binding affinity [21,22,23,24]. In this study, DMS data from Wu et al. [24] and Wang et al. [23] were used to construct empirical fitness landscapes for a binding domain of the Streptococcal bacteria Protein G (GB1) and an epitope region of the human influenza H3N2 neuraminidase (NA) antigen, respectively.

### 1.3. Evolutionary Algorithms

The availability of both mathematical and empirically generated fitness landscapes has generated interest in evolutionary algorithms as a means of simulating evolution across these spaces. Studies have evaluated the use of objective and novelty search algorithms, with performance found to be dependent on the physical characteristics of the space [17,25,26,27,28]. The failure of the goal-oriented objective search algorithms in increasingly deceptive (rugged) environments identified the need for an alternative approach, the novelty search [26]. Novelty searches, in contrast, abandon the objective altogether. Rather than seeking a fitness-oriented goal, novelty searches seek out spaces within the landscape that have not been explored. By rewarding behaviors that explore novel spaces, an escape opportunity is provided when local maxima are reached. Although it has been demonstrated that novelty-based searches can outperform objective searches in deceptive landscapes, they fail in large and unconstrained spaces [28]. To resolve these constraints on the two search methods, combinations of objective and novelty searches have been explored. By retaining the goal-oriented approach of objective searches and combining it with the navigational flexibility of the novelty search, the combination has been shown to outperform objective or novelty approaches alone in unconstrained, deceptive environments [28]. Whereas previous studies have evaluated the performance of objective, novelty, and combination searches in non-biological settings, our study evaluates the different search methods on both simulated and empirically generated fitness landscapes.

## 2. Algorithms and Methods

### 2.1. Simulated Fitness Landscapes Based on NK Model

The NK model was used to generate simulated fitness landscapes, as originally described by Kauffman and Weinberger [12]. The NK landscapes were generated using *N* = 10-bit binary sequences, with *A* = 2 states (0 or 1) and *K* = 0 to *N* − 1 epistatic interactions per position. Each position on the sequence was assigned a random value of fitness to contribute, and each of the 2*^N^* possible combinations generated by the model are representative of the different possible genotypes. Simulated landscapes with higher *K* values produced more rugged, or deceptive, landscapes with a greater number of local peaks, while using smaller *K* values produced smoother landscapes with less peaks. The NK model is commonly used to explore evolution involving complex epistatic interactions among mutation sites [29]. The NK-family landscapes are also a standard model of a complex system, represented as a directed graph consisting of interacting nodes [30,31].

The Python script used to generate NK landscapes was derived from a Github resource (https://github.com/Mac13kW/NK_model/blob/master/1_landscape_creation.py, accessed on 1 November 2022), with the following modifications. First, we normalized the fitness values to be within 0 and 1 (inclusive). Second, we added an option for generating exponentially distributed fitness values, as is often the case in empirical fitness landscapes [32]. Fitness values were indeed exponentially distributed in the two empirical landscapes employed in the present study (see below), where most genotype variants showed low fitness values (deleterious relative to the wildtype) with a long tail of high-fitness genotype variants (beneficial relative to the wildtype).

### 2.2. Empirical Landscapes

#### 2.2.1. GB1 4-Epitope Site Landscape

Streptococcal Protein G binds antibodies, and a combinatorically complete fitness landscape, have been derived at four epitope sites (V39, D40, G41, and V54) at its 56-amino acid B1 domain [24]. Fitness values (*w*) for a total of 20^4^ = 160,000 4-aa haplotypes—correlated with organismal fitness—were defined as folding stability and binding affinity to the conserved Fc region of the IgG antibody molecules, relative to those of the wildtype haplotype (“VDGV”, *w* = 1). Fitness values of a small portion (6.6%) of haplotypes absent in experimental data were computationally imputed according to a regression model [24]. We downloaded the experimentally profiled dataset (https://doi.org/10.7554/eLife.16965.024, accessed on 1 June 2022) and the imputed dataset (https://doi.org/10.7554/eLife.16965.025, accessed on 1 June 2022) and combined the two into a single file (available at https://github.com/weigangq/nov-search.git, accessed on 1 June 2022, “data/GB1-supp1.csv”). To simplify the visualization of the GB1 landscape, we extracted a subset landscape consisting of 625 haplotypes, each of which was a unique combination of 5 amino acids at each site. The five amino acids included the wildtype residue and the top four most frequent residues involved in the high-order epistasis [24]. Specifically, the residues of the haplotypes on the sub-landscape consisted of Val, Trp, Tyr, Phe, and Gly residues at site 39; Asp, Leu, Gln, Glu, and Thr residues at site 40; Gly, Leu, Cys, Phe, and Tyr residues at site 41; and Val, Pro, Ser, Tyr, and His residues at site 54.

#### 2.2.2. NA 7-Variable SITE Landscape

The molecular surface of the influenza virus consists of the two major antigens hemagglutinin (HA) and neuraminidase (NA), both of which are targets of vaccination [3]. A fitness landscape of seven highly variable epitope sites on NA of human influenza virus H3N2 has been derived experimentally with high-throughput mutagenesis and next-generation sequencing [23]. The seven highly antigenic sites were structurally proximate, and the landscape consisted of 864 complete combinations of haplotypes made up of the following list of amino acid residues: Asn and Lys at site 328; Asn, Ser, and Asp at site 329; Lys, Glu, and Arg at site 344; Asn, Ser, and Gly at site 367; Lys and Glu at site 368; Lys, Glu, Asp, and Thr at 369; and Leu and Ser at residue 370. Viral fitness was measured for six genetic backgrounds represented by six natural viral isolates (HK68, Bei89, Mos99, Vic11, and HK19), with the wildtype fitness defined as *w* = 0, deleterious variants as *w* < 0, and beneficial variants as *w* > 0. Fitness values of the same 7-residue haplotypes were highly correlated among the genetic backgrounds [23]. In the present study, we used the fitness landscape of the HK19 background, which was downloaded from the Github repository (https://github.com/Wangyiquan95/NA_EPI/blob/master/result/NA_compile_results.tsv, accessed on 1 June 2022).

### 2.3. Characterization of Fitness Landscapes

Simulated NK landscapes consisting of binary strings were characterized by the size of the search space (the number of strings, *2^N^*), the number of local and global fitness peaks (*N_max_*), and a measure of ruggedness (*r*/*s*) [29]. A Python script (“land-stats.py”) was developed to obtain values of these landscape characteristics, based on the Github resource (https://github.com/song88180/fitness-landscape-error/blob/master/utils/utils.py, accessed on 1 June 2022).

Computationally, the NK and empirical fitness landscapes were implemented as directed graphs, with nodes representing the genotypes. Edges connect two genotypes separated by a single mutation, with the direction pointing from the low- to the high-fitness genotypes. The Python module NetworkX was used to implement, analyze, and render the landscape graphs [33]. In the graph implementation of a fitness landscape, fitness peaks were identified as nodes with only *in_degree* edges and fitness valleys as nodes with only *out_degree* edges. Further, the graph implementation allowed for characterization of landscapes by local optima network (LON) analysis [34]. For example, the basin of attraction of a fitness peak is defined as the number of starting nodes, from which an adaptive walk, following the greatest fitness gain at each mutation step, would reach the fitness peak [34]. Fitness peaks with a larger basin of attraction are easier to reach by the steepest fitness climb (greedy algorithm).

### 2.4. Evolutionary Walks

Evolutionary walks on fitness landscapes are examples of evolutionary algorithms (EAs), which imitate the naturally open-ended and creative process of biological evolution to search for solutions to combinatorial optimization problems [35,36]. Evolutionary walks were performed on simulated and empirical fitness landscapes (Figure 1). Each walk started with a population (e.g., *n* = 100) of randomly derived binary strings or amino-acid haplotypes and ended when the fitness peak was reached or when a predefined generation limit (e.g., *g_max_* = 100) was reached. During each generation, a mutation was introduced to a randomly chosen site at each haplotype with a Poisson probability (at an average rate of, e.g., *m* = 1 per haplotype per generation). Fitness (or novelty) values of mutated haplotypes were evaluated according to the fitness landscape (or a sparsity function, see below), and the top valued haplotypes (e.g., *e* = 10 “elites”) were chosen to seed the next generation of the population. Each replicates 1/10 × *n* times, resulting in a new (and more fit) population of *n* individuals.

Evolutionary walks, which use a population as the base unit, differ from adaptive walks based on individuals. Individual-based adaptive walks do not construct a population of haplotypes, introduce mutations, or select top-scored individuals for the next generation. For example, the authors simulated adaptive walks of individual haplotypes on the GB1 landscape according to three algorithms [24]. With the first algorithm (the “greedy” model), at the selection step, a single haplotype with the highest fitness score was selected, and the walk stopped when there were no haplotypes with a greater fitness score to select. With the other two algorithms, at each selection step, an individual was chosen non-deterministically, based on a probability according to its fitness (the “correlated fixation” model) or on the equal probability (the “equal fixation” model). The individual-based adaptive walks were used to quantify the classes of adaptive paths (directly accessible, indirectly accessible, and inaccessible paths) and their length distributions in a fitness landscape.

Following the example of simulated evolutionary walks on mazes by a self-navigating robot, where a successful walk was defined as a path ending at the maze exit [28], we implemented the following three search algorithms for finding the global peak in a fitness landscape.

#### 2.4.1. Fitness-Seeking Walks

The objective evolutionary algorithm adapts a sequence over generations via selection based on fitness scores. First, a fitness landscape is generated using the NK model (for simulated landscapes) or deep mutational scan data (for empirical landscapes). A starting point on the landscape is selected at random, and a population of *n* = 100 is generated from the respective starting sequence by replicating 100 times and applying an average of *m* = 1 random mutation for each new sequence. The top 10 sequences with the highest fitness values, the “elites”, are selected, and used to regenerate the population. This continues for a total of *g_max_* = 100 generations, or until the global maximum is reached, in which case the search is terminated.

#### 2.4.2. Novelty-Seeking Walks

The novelty evolutionary algorithm adapts a sequence over generations via selection based on novelty scores. The same steps are executed as in the objective algorithm, with the only difference being selection on novelty rather than fitness. To calculate the novelty score, first a behavior was defined [28]. Here, the behavior of a haplotype was defined as the coordinate of its position in a multidimensional geometric space. For the *N*-bit binary strings in the NK landscape, the coordinate of a string was the set $x={\left\{0,1\right\}}^{N}$. For the *N*-amino acid sequences in empirical landscapes, a haplotype belonged to the set ${\left\{A,\;C,\;D,\ldots \right\}}^{N}$, where *A*, *C*, *D*, … were all the possible amino acid residues at each site. The coordinate of a haplotype in the *N*-dimensional space was defined by an array of physiochemistry values consisting of the polarity (*pol*) [37], hydrophobicity (*hydro*) [38], and isoelectric point (*iep*) [39] for an amino acid residue obtained from the AAindex database [40]: $x={\left\{\left(A_{pol},\;A_{hydro},\;A_{iep}\right),\left(D_{pol},\;D_{hydro},\;D_{iep}\right),\;\ldots \;\right\}}^{N}$. Second, *k* nearest neighbors were found for each individual in the population and archived. The sparsity score for an individual, *s(x)*, was then calculated by finding the average *N*-dimensional distance to the *k* nearest neighbors and those previously archived (*u_i_*). The sparsity score of a binary string was the average Hamming distance to its *k* neighbors: $s\left(x\right)=\sum_{i=0}^{k}Hamming\left(x,u_{i}\right)$, following the notation of [28]. For an amino acid haplotype, its sparsity score was the average Euclidean distance to the *k* nearest neighbors: $s\left(x\right)=\frac{1}{k}\sum_{i=0}^{k}\sqrt{\sum_{N}\left(\delta_{pol}{}^{2}+\delta_{hydro}{}^{2}+\delta_{iep}{}^{2}\right)}$, where *δ* was the difference in a physicochemical measure between the haplotype and a neighbor. In both cases, the larger the sparsity score, the more novel a genotype was. Third, the top 10 sequences with the highest sparsity scores were selected as the most novel. The archive was updated for each round of selection and maintained at either an unrestricted size or a fixed size by replacing the oldest values with the newest values [25,27]. As such, the sparsity of the neighboring region surrounding a string was quantified as the average distances of the string to its 10 closest neighbors.

#### 2.4.3. Fitness–Novelty Hybrid Walks

The combination search algorithm follows the same steps as the objective and novelty searches alone, apart from the selection step. In the combination algorithm, the fitness and novelty scores for each genotype in the population are normalized, then combined with equal weights (*p =* 0.5 each): $score\left(x\right)=\left(1-p\right)w_{x}+ps_{x}$, where *w_x_* and *s_x_* were the normalized (between 0 and 1) fitness and sparsity values, respectively. Selection of a new top 10 set of elites was based on the combined score.

### 2.5. Landscape Visualization and Data Analysis

Our computational tools automate the visualization scheme of representing fitness landscapes as directed graphs [11,24]. Fitness landscapes were visualized with the multipartite layout, where genotypes were arranged according to their mutational distances from the wildtype.

For an NK fitness landscape, the wildtype node consisted of an *N*-bit string of all 0′s, and the fully mutated node consisted of an *N*-bit string of all 1′s. At a Hamming distance of *d* from the wildtype, there were a total of $\left(\begin{array}{c} N\\ d \end{array}\right)$ genotypes. The total number of genotypes in the entire landscape was thus $\sum_{d}\left(\begin{array}{c} N\\ d \end{array}\right)=2^{N}$. Each genotype was attached by *N* incoming or outgoing edges. The total number of edges in the landscape was $\sum_{d}\left(N-d\right)\left(\begin{array}{c} N\\ d \end{array}\right)=N2^{N-1}$. The fitness peaks were identified as nodes consisting of only incoming edges, and the fitness valleys were nodes consisting of only outgoing edges.

For a combinatorically complete empirical landscape, the wildtype node consisted of an *L*-length peptide sequence, with the set of potential amino acid residues at a position *i* being $S_{i}$, e.g., $S_{2}=\left\{Ala,\;Cys,\;Met\right\}$. The total number of unique amino acid residues at a position *i* was thus $s_{i}=\left|S_{i}\right|$. At a distance of *m* mutations from the wildtype, there were a total of $\sum_{1}^{\left(\begin{array}{c} L\\ m \end{array}\right)}\prod_{i}^{m}s_{i}$ genotypes, which simplifies to $\left(\begin{array}{c} L\\ m \end{array}\right)19^{m}$ in the case that each site may mutate to all 19 alternative amino acid residues.

## 3. Results

### 3.1. Simulated Landscapes

Simulated NK landscapes were generated with binary sequences of *N* = 10 and *K* values ranging from 0 to *N*-1, producing networks with *2^N^* = 1024 nodes representing different genotypes. By adjusting the *K* value in the model, we were able to tune the ruggedness of the landscape. Increasing *K* results in a greater number of peaks in the landscape and consequently decreases the basin of attraction of the global peak, thus creating varied ruggedness in the environments to test the evolutionary algorithms (Table 1). For *K* = 2, the fitness landscape contains a total of 8 peaks and 6 valleys (Figure 2, left). As shown in the subgraph for the global peak (Figure 2, top right) and a random valley (Figure 2, bottom right), each node has a total of 10 1-degree edges.

### 3.2. Performances on Simulated Landscapes

Adaptive evolution of microbial populations was simulated on our NK landscapes by executing our 3 evolutionary algorithms with populations of *n* = 100 binary strings (Figure 3). Ruggedness of the landscape was tuned for different trials by adjusting the *K* value from *K* = 0 to *K* = 9. Evolutionary walks were repeated 50 times for each algorithm and at each level of landscape complexity (*K*). The objective (fitness seeking) search performed well on smoother landscapes, with performance diminishing considerably when tested on more deceptive landscapes (Figure 3A, top). Each successive generation increased in fitness monotonically, but frequently got stuck at local fitness peaks as the landscape complexity increased (Figure 3B, top). Landscape complexity had no effect on the novelty search, with the global peak only being reached on any given trial by chance (Figure 3A, middle). Generational fitness levels showed the highest degree of fluctuation with the novelty algorithm at all levels of complexity (Figure 3B, middle). The hybrid search was effective in finding the global peak on all landscapes, although the number of generations to reach the global fitness peak increased with landscape complexity (Figure 3A, bottom). As observed with the objective search, fitness levels for each successive generation trended upward using the hybrid search, but fluctuations were consistently observed on more complex landscapes (Figure 3B, bottom).

### 3.3. Empirical Landscapes and Performance Measures

#### 3.3.1. Streptococcal GB1

The experimentally generated full fitness landscape for the 4 epitope sites of GB1 consists of 160,000 haplotypes [24]. This complete, highly rugged landscape has a peak-to-haplotype ratio of 1:25, with the basin of attraction for the global peak calculated at 0.36% (Table 1). A subset landscape was constructed using the 625 haplotypes, composed of the top 5 amino-acids at each of 4 epitope sites with the strongest high-order epistasis (Figure 4).

Adaptive walks were simulated on the full GB1 landscape for each of our three evolutionary search algorithms (Figure 5A). The hybrid search outperformed both the objective and novelty searches alone, finding the global fitness peak in less than 25 generations on average. The few trials in which the objective search found the global peak were due to chance, with initial populations likely starting in very close proximity to the global peak. Due to the vastness of the genotypic space (160,000 haplotypes), the novelty search failed on every simulation. Results from three separate adaptive walks (chosen randomly) for each of the three search algorithms reveal expected patterns of generational fitness gains (Figure 5B). The fitness search (Figure 5B, top panel) shows a steady increase in fitness over generations, with plateaus signifying walks between haplotypes with either close fitness values or the inability of the search to escape a haplotype. Fitness gains across generations for the novelty search (Figure 5B, middle panel) fluctuated slightly, but never increased significantly. Adaptive walks using the hybrid search (Figure 5B, bottom panel) showed a fitness increase from generation to generation, with small fluctuations. These fluctuations, where fitness actually decreases from one generation to the next, represent instances where fitness may need to decrease to escape a genotype, giving subsequent generations access to higher fitness peaks.

#### 3.3.2. Human Influenza NA

A fitness landscape for human influenza NA was experimentally generated by measuring replication fitness via high-throughput combinatorial mutagenesis at seven structurally proximate and highly variable epitope sites on the NA protein on six genetic backgrounds (Figure 6) [23]. The plot shows the complete landscape of 864 haplotypes on the HK19 genetic background, consisting of the wildtype (“KSENETS”, *w* = 1) and the global fitness peak (“KDRSETS”, *w* = 8.80). The landscape is complex, with a 1:22 peak to haplotype ratio and the basin of attraction for the global peak calculated at 0.46% (Table 1).

Adaptive walks were simulated on the NA landscape for all three evolutionary algorithms (Figure 7A). Similar to what we observed for the GB1 landscape, the hybrid search outperformed both the objective and novelty searches and successfully navigated to the global peak on all runs. The fitness and novelty search results were consistent with the adaptive walks simulated on the NK landscapes of similar size and complexity. Compared to the GB1 landscape, the smaller genotypic space on the NA landscape seemed to have benefitted the fitness search more than the novelty search, as the mean generations to the global peak was less for the former. The population fitness over generations for each of the three search algorithms were also consistent with the adaptive walks observed for the GB1 and NK landscapes (Figure 7B). The hybrid search showed population fitness increasing over generations, with occasional small fluctuations (Figure 7B, bottom). The fitness search showed steady increases in population fitness across generations, with a plateau representing local peaks that could not be escaped (Figure 7B, top). The novelty search showed the greatest amount of fluctuation over generations, as would be expected for a search lacking an objective goal (Figure 7B, middle).

## 4. Discussion

### 4.1. Evolutionary and Non-Evolutionary Walks

In the present study, we explored evolutionary mechanisms of antigenic diversification in microbial pathogen populations by: (1) constructing theoretical and empirical fitness landscapes representing antigen variants, (2) characterizing fitness landscapes using local optima networks (LONs), and (3) performing simulated walks with three evolutionary algorithms. We caution, however, that the fitness of a pathogen strain carrying a particular antigen variant may not be directly correlated with phenotypic measurements, such as receptor binding affinity [41]. Indeed, fitness values of variants of a protein should ideally be evaluated at multiple levels of biological interactions, including intra-molecular, metabolic, cellular levels, and beyond [20].

Nonetheless, fitness landscapes are frequently used for the study of evolutionary predictability, with the premise that sequence evolution most likely follows accessible paths on a landscape [11,42]. For example, on the empirically defined GB1 landscape, the authors performed adaptive walks, with each interim step determined by the steepest fitness gain (“greedy” model), by a probability correlated with fitness gains (“correlated fixation” model), or by an equal probability (“equal fixation” model) [24]. These simulated adaptive walks, which trace evolutionary paths from a single individual haplotype as the starting point toward a local fitness peak as the terminating point, were not strictly evolutionary algorithms. Evolutionary computation (EC; e.g., with genetic algorithms) emulates natural evolution by constructing and maintaining a finite-sized population of individuals during simulated walks, thus introducing the mechanism of genetic drift (Figure 1) [43]. In addition, evolutionary computation introduces random mutations (and, optionally, recombination) at each generation. Further stochasticity is introduced during the selection step, by which a group of high-fitness individuals (“elites”) are chosen to populate the next generation. In sum, evolutionary algorithms (Figure 1), extensively used in robotics and artificial intelligence [28,35,36], are novel algorithms, but paradoxically thus far, are rarely used for exploring evolutionary dynamics and predicting evolutionary trajectories using protein fitness landscapes. We show below that the application of evolutionary algorithms to a protein fitness landscape diminishes the predictability of evolutionary paths while enhancing the predictability of high-fitness antigen variants.

### 4.2. Biological Factors Facilitating Evolution of Novel Antigens

The efficiency and consistency of the hybrid fitness-and-novelty-seeking algorithms to evolve high-fitness variants, even on the highly complex fitness landscapes (Figure 4, Figure 5, Figure 6 and Figure 7), suggests that landscape topology—however rugged—may not impose severe constraints on antigen diversification in natural pathogen populations. Traditionally, by geometrically representing the number, degree, and types of epistasis, fitness landscapes have become a fundamental model for predicting the dynamics and trajectory of protein sequence evolution [11,29,42]. However, algorithms of adaptive walks may influence how strongly a fitness landscape constrains protein sequence evolution as much as the landscape topology itself. For example, when the greedy algorithm was applied to the GB1 landscape with the wildtype as the starting point, ~50% of high-fitness peaks were inaccessible, ~30% indirectly accessible, and only ~20% directly accessible (Figure 4F in [24]). In contrast, on the same fitness landscape, all fitness peaks (including the highest one) were quickly accessible with the use of hybrid search algorithms (Figure 5). If antigen diversification in natural pathogen populations follows a process close to the hybrid fitness-and-novelty walks, evolutionary paths would not be constrained by landscape topology, making evolutionary dynamics and trajectories less predictable than those suggested by fitness-based adaptive walks. Meanwhile, since virtually all fitness peaks are reachable by a combination of fitness- and novelty-seeking walks, the evolutionary *outcomes* of antigen diversification—represented by these fitness peaks—would be almost deterministically predictable once an empirical fitness landscape is defined.

Biological processes facilitating novelty seeking in natural pathogen populations include the genetic processes of recombination and hypermutability. Homologous recombination introduces groups of mutations in a single generation, carrying populations over large distances on a fitness landscape. Recombination among coexisting strains is common in natural populations of microbial pathogens [44,45,46]. Strains, multilocus linkage groups, or genomic clusters present in natural pathogen populations are often consequences of natural selection or rapid clonal expansion, and not due to a lack of recombination [4,8,46]. Further, microbial pathogen genomes encode specific hyper-mutation and hyper-recombination mechanisms at antigen loci, such as the antigen variability *vls* system in Lyme disease pathogens (*Borreliella* species), genes encoding the variant glycoproteins (VSGs) in the African trypanosome (*Trypanosoma brucei*), and genes conferring adhesion, toxin, and antigenic variabilities in pathogenic *Neisseria* species [5,47,48]. In microbial populations without recombination, such as those in long-term experimental evolution studies that provide a defined and controlled environment, hypermutants periodically appear due to mutations in DNA repair genes [49,50]. In sum, genetic mechanisms, such as homologous recombination among genomes, antigen-specific hyper-recombination, and mutator phenotypes, facilitate the novelty-seeking behavior of microbial pathogen populations by quickly generating a vast number of antigen variants that cover a large span of their fitness landscape.

Diversifying natural selection targeting pathogen antigen loci preferentially retains novel antigen variants, thus promoting novelty-seeking walks by microbial pathogens. In Lyme disease pathogens, the outer-surface protein C (*ospC*) locus is the most variable locus in the genome as a result of negative, frequency-dependent selection, which favors rare antigen variants [8,51]. Similarly, diversifying selection drives rapid sequence diversification among the *vls* paralogs within and between closely related *B. burgdorferi* genomes [52,53]. Similarly driven by the fixation of novel amino-acid sequences, cell wall glycoproteins in yeasts evolve quickly and adaptively [54].

Further, population and epidemiological processes facilitate the evolution of novel antigen variants through genetic drift. For viruses and vector- or host-dependent bacterial pathogens, every transmission among hosts creates a population bottleneck, by which only a fraction of infectious agents succeed in reaching the next host. Long-distance pathogen dispersal creates founder effects, by which only a small subset of the original pathogen populations reach a new destination. For a novel human pathogen like the SARS-CoV-2 coronavirus, fast dispersal to multiple locations during a pandemic allowed parallel independent evolution across the world, with each local outbreak exploiting a novel evolutionary trajectory [22,55]. In all the above cases, genetic drift weakens natural selection and facilitates the emergence of novel high-fitness pathogen strains by generating and maintaining low-fitness transitory haplotypes.

### 4.3. Variant Predictability and Design of Broadly Protective Vaccines

Despite the diminished impact of fitness landscapes in constraining and predicting evolutionary dynamics and trajectories, evolutionary outcomes represented by the fitness peaks are more predictable under fitness-and-novelty-seeking walks. As such, the fitness landscape of a pathogen antigen remains critically important for predicting the emergence of high-fitness antigen variants (e.g., variants of interest, or VOIs). For example, the sizes of basins of attraction of individual fitness peaks are correlated with haplotype fitness values and could be used to quantify the frequency distribution or likelihood of future antigen variants [24,56].

Intriguingly, characteristics of fitness landscapes can aid in the design of broadly protective and potentially escape-proof vaccines against a microbial pathogen [1,2]. Previously, we used evolutionary algorithms to develop synthetic peptides that were broadly cross-reactive with natural outer-surface protein C (OspC) variants in Lyme disease pathogen populations [57]. Others have taken a similar approach of designing broadly protective vaccines based on phylogenetic consensus and root sequences [58,59,60]. These evolution-inspired designs of escape-proof vaccines generate synthetic peptides close to the center of phylogenetic trees, and they are consistent with models of immune selection asserting a positive correlation between sequence and immunological distances [7].

To illustrate, we identified a total of 6409 fitness peaks on the full GB1 landscape, consisting of 160,000 haplotypes. The top 25 peaks with the largest basins of attraction encompassed 51.6% of the total fitness landscape (Figure 8). The log-scaled basin size is positively correlated with fitness (*R*^2^ = 0.48, *p* < 2 × 10^−16^) (Figure 8, inset), consistent with the findings of local optima network (LON) analysis of simulated and protein landscapes [34,56]. These peaks represent antigen variants most likely to emerge from the pathogen population. Evolutionary centroids that are equidistant to these top-basin variants are good candidates for potentially escape-proof vaccines [7,57,58,59].

## 5. Conclusions

Facilitated by large and rapid genetic changes brought by recombination, hypermutability, genetic drift, and diversifying natural selection, novelty search promotes antigenic diversity in natural microbial pathogen populations. Simulated evolution with a combination of fitness- and novelty-seeking algorithms provides a way to predict the rise of high-fitness antigen variants given an empirically derived antigen fitness landscape. Evolutionary centroids of high-fitness variants are promising candidates for broadly protective, escape-proof vaccines against a diversifying microbial pathogen.

## Figures and Tables

**Figure 1 pathogens-12-00388-f001:**
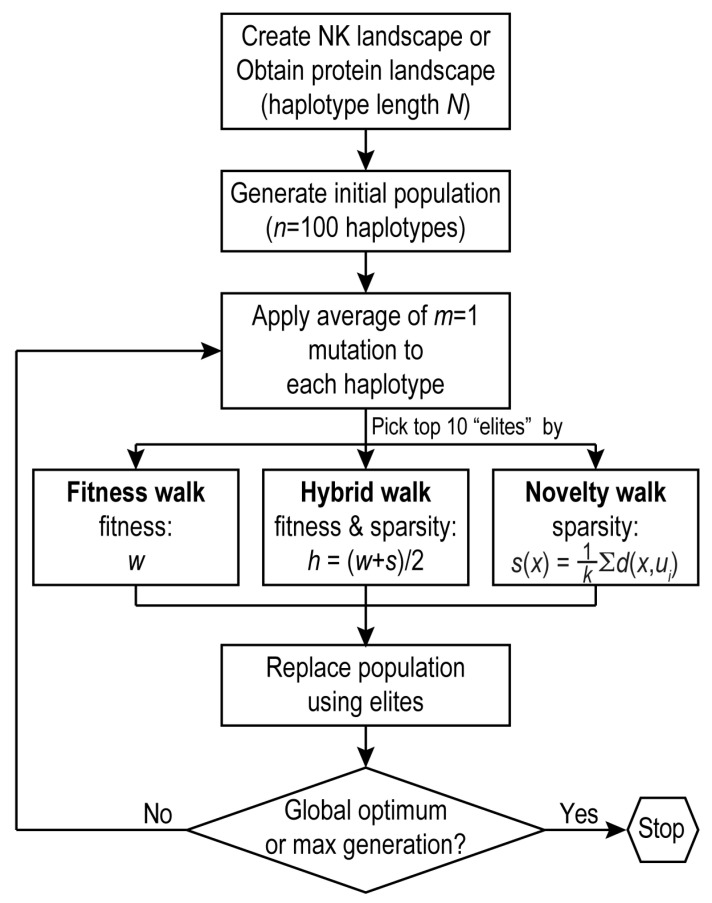
Evolutionary search algorithms. For evolution on a simulated landscape, an initial population of antigen variants (with *n* individuals) were randomly generated. Elites were defined by either the fitness score (fitness walk, (**left**)), the novelty score (novelty walk, (**right**)), or a linear combination of fitness and novelty scores (hybrid walk, (**middle**)). Novelty of a haplotype (*x*) is measured by sparsity, calculated as average distance to its *k* nearest neighbors (*u*) (see text for details).

**Figure 2 pathogens-12-00388-f002:**
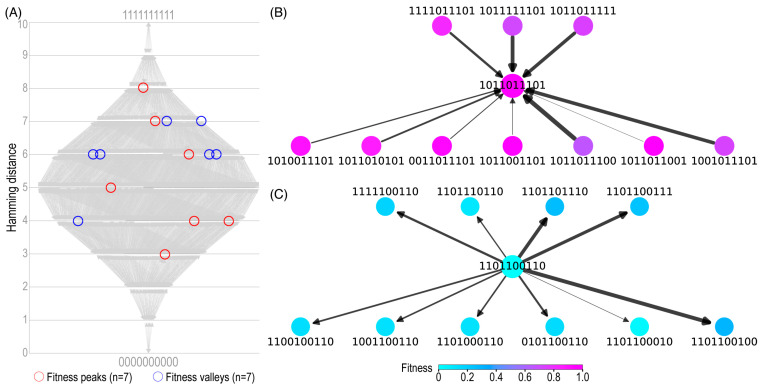
A simulated NK landscape. (**A**) Multipartite renderings of a simulated fitness landscape using the NK model of mutation interactions (*N* =10, *K* = 2). Nodes (*n* = 1024) are genotypes represented by 10-bit binary strings. Edges (*n* = 5120 in total) connect two genotypes separated by a Hamming distance of *d* = 1. (**B**) A subgraph showing nodes separated from the global fitness peak node (center, *w* = 1) by 1 degree. (**C**) A subgraph showing nodes separated from the global fitness valley (center, *w* = 0) by 1 degree. In the two subgraphs, edge widths are scaled proportionally with fitness changes and nodes are shaded by fitness values according to the color map.

**Figure 3 pathogens-12-00388-f003:**
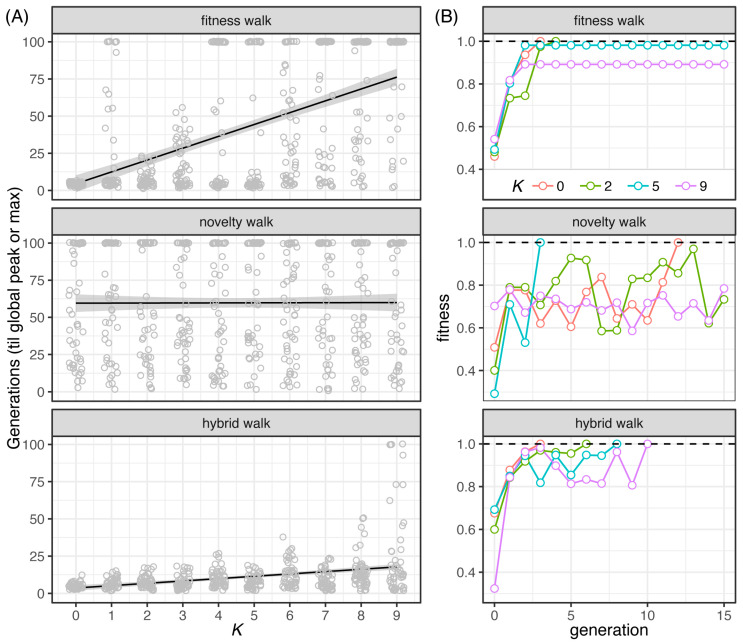
Evolutionary walks on NK landscapes. (**A**) Simulation of adaptive evolution of a microbial population on ten NK landscapes with the length of binary strings *N* = 10 and the number of interacting sites ranging from *K* = 0 to *K =* 9 (*x*-axis). Populations evolved according to three evolutionary algorithms (Figure 1): fitness-seeking (top panel), novelty-seeking (middle panel), and hybrid walks (bottom panel). Strings mutated at an average rate of one bit flip per haplotype per generation. Each point represents the number of generations (*y*-axis) either when the population reached the global fitness peak (*g* < 100) or when the simulation reached the generation limit (*g* = 100). (**B**) Fitness values (*y*-axis) of the fittest individuals at each generation (*x*-axis) were plotted for populations evolving under three search algorithms at four complexity levels (*K* = 0, 2, 5, and 10). Note that novelty of a binary string was defined as the unique position of the string in an *N*-dimension hypercube (see Algorithms and Methods).

**Figure 4 pathogens-12-00388-f004:**
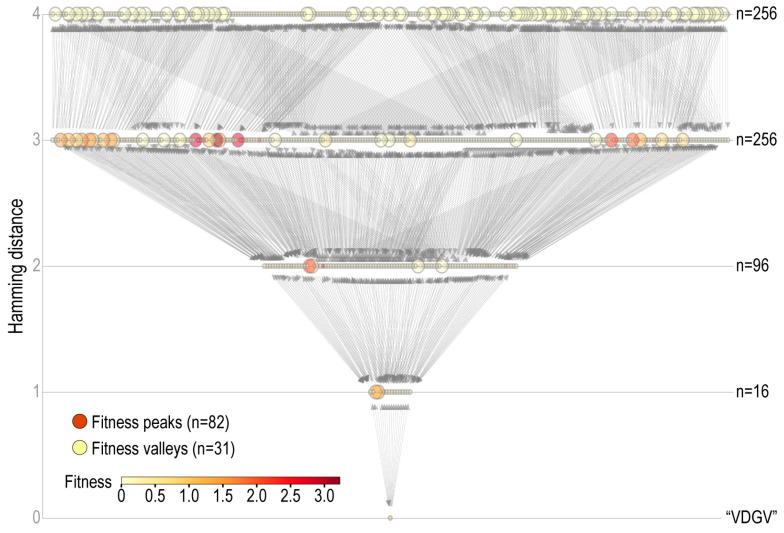
Subset of GB1 fitness landscape. Nodes (*n* = 625) represent haplotypes and are colored according to the fitness value gradient. Larger-sized nodes indicate fitness peaks and valleys. The full GB1 landscape (not shown) consists of 20^4^ = 160,000 haplotypes including the wildtype (“VDGV”, *w* = 1), the global peak (“AHCA”, *w* = 9.91), 6409 fitness peaks, and 37,089 fitness valleys.

**Figure 5 pathogens-12-00388-f005:**
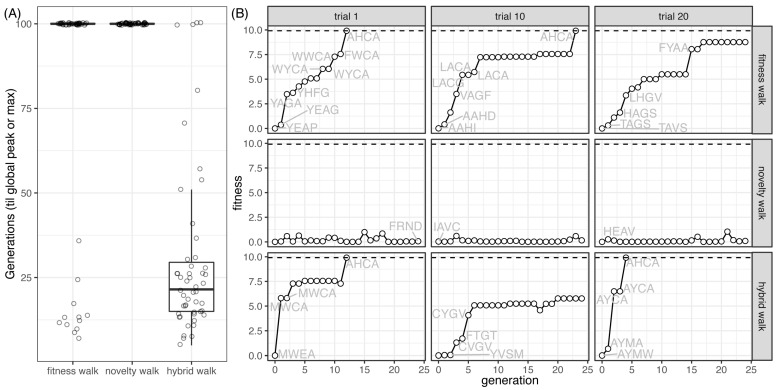
Evolutionary walks on the full GB1 landscape. (**A**) Number of generations (*y*-axis) when a population (represented by a data point) of 100 GB1 4-amino acid peptides reached either the global peak (“AHCA”) or the maximum generation limit (*g* = 100) for the three evolutionary walks (*x*-axis). Evolutionary walks were repeated 50 times for each algorithm. (**B**) Fitness values (*y*-axis) of the fittest individual (genotype labeled) over generations (*x*-axis) from three randomly chosen independent walks (columns) for each search algorithm (rows).

**Figure 6 pathogens-12-00388-f006:**
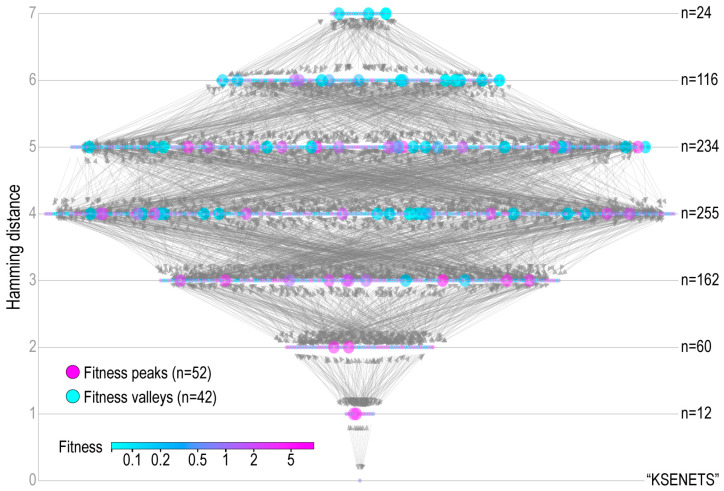
NA fitness landscape for 7 epitope sites for the HK19 genetic background. Nodes (*n* = 864) are colored according to the fitness value gradient. Larger node size indicates fitness peaks and valleys. Wildtype epitope sequence on bottom level (hamming distance = 0).

**Figure 7 pathogens-12-00388-f007:**
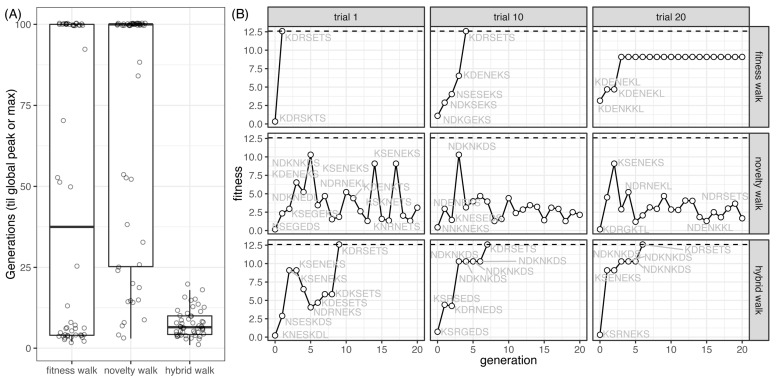
Evolutionary walks on the NA landscape. (**A**) Number of generations (*y*-axis) when a population (represented by a data point) of 100 NA 7-amino acid epitopes reached either the global peak (“KDRSETS”) or the maximum generation limit (*g* = 100) for the three evolutionary walks (*x*-axis). Evolutionary walks were repeated 50 times for each algorithm. (**B**) Fitness values (*y*-axis) of the fittest individual (genotype labeled) over generations (*x*-axis) from three randomly chosen independent walks (columns) for each search algorithm (rows).

**Figure 8 pathogens-12-00388-f008:**
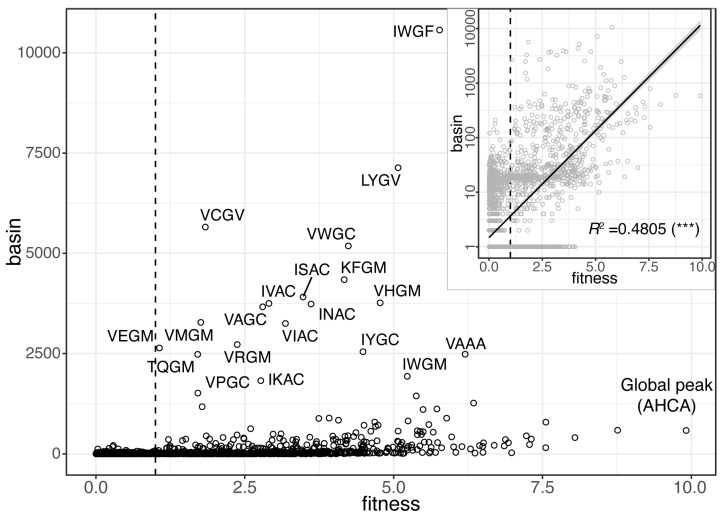
Local optimal network (LON) analysis. A plot of 6409 fitness peaks on the full GB1 landscape, with the position of a peak (a dot) represented by its fitness (*x*-axis) and size of basin of attraction (*y*-axis). A dashed line marks the fitness value of the wildtype (“VDGV”, *w* = 1). Top 20 haplotypes with basin size >1500 are labeled. Note that the global fitness peak (“AHCA”, *w* = 9.91) is not part of the top basin peaks. (inset) Plot of log-scaled basin size and fitness (*** *p* < 2 × 10^−16^).

**Table 1 pathogens-12-00388-t001:** Landscape characterization.

Landscape	Hap Length	#Haps *^a^*	#Peaks (Peak/Hap Ratio *^b^*)	Fitness Distribution *^c^* {min, max}	Ruggedness (*r*/*s* Ratio *^d^*)	Basin of Attraction (*b*) *^e^*
NK (*K* = 0)	10	1024	1 (1:1024)	*N*{0, 1}	3 × 10^−6^	100%
NK (*K* = 1)	10	1024	4 (1:256)	*N*{0, 1}	2.63	35.1%
NK (*K* = 2)	10	1024	7 (1:146)	*N*{0, 1}	1.79	26.4%
NK (*K* = 3)	10	1024	15 (1:68)	*N*{0, 1}	4.79	12.4%
NK (*K* = 4)	10	1024	22 (1:47)	*N*{0, 1}	4.24	12.4%
NK (*K* = 5)	10	1024	28 (1:37)	*N*{0, 1}	6.10	10.4%
NK (*K* = 6)	10	1024	48 (1:21)	*N*{0, 1}	8.19	8.11%
NK (*K* = 7)	10	1024	59 (1:17)	*N*{0, 1}	10.3	6.45%
NK (*K* = 8)	10	1024	73 (1:14)	*N*{0, 1}	15.2	1.95%
NK (*K* = 9)	10	1024	101 (1:10)	*N*{0, 1}	22.2	1.37%
GB1	4	160,000	6409 (1:25)	*E*{0, 9.91}	*n.a.*	0.36%
NA	7	864	40 (1:22)	*E*{0.06, 8.81}	*n.a.*	0.46%

*^a^* Number of haplotypes: a measure of search space. The larger the number of haplotypes, the more difficult it is for a novelty-seeking walk to find the global fitness peak. *^b^* Peak-to-hap ratio: a measure of landscape deceptiveness. The higher the ratio, the more likely it is for a fitness-seeking walk to be trapped on a local fitness peak. *^c^ N*: normal distribution (truncated between 0 and 1). *E*: exponential distribution. *^d^ r*/*s* ratio: A measure of binary-string landscape ruggedness based on linear regression parameters on additive model [29]. *n.a*., not available. *^e^* Size of basin of attraction (of the global peak): measured as the percentage of haplotypes starting from which a steepest fitness climb will reach the global peak [34].

## Data Availability

All datasets and computer scripts (in BASH, PYTHON, and R) are publicly available at the Github repository: https://github.com/weigangq/nov-search.git, accessed on 1 June 2022.

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
