# Peer review of "Novelty Search Promotes Antigenic Diversity in Microbial Pathogens"

_pathogens, 2023, doi:10.3390/pathogens12030388_

Round 1

Reviewer 1 Report

Dear Authors,

in my opinion prepared by you manuscript is very interesting both in practical, as well in cognitive context and contributes a lot to microbiology, evolutionism, molecular phylogeny, immunology of host-pathogen interactions and finally to mathematical models to analyze the rate of evolution. The research conducted by you will certainly draw the attention of the scientific community on the possible alternative strategies to study host-pathogen coevolution. All the figures and tables are appropriate for this type of article. In general, the paper has a logical flow. The abstract well correspond with the main aspects of the work. Nevertheless, as a reviewer I am obligated to pay attention even to less important shortcomings of this work and all mentioned below comments should be carefully considered.

Abstract

Abstract counts 244 words and according to Instructions for Authors should be shortened to 200 words maximum.

References

Another issue is related with references, namely in the text reference numbers should be placed in square brackets [ ].

Page 2, line 31

According to the way of writing adopted by the authors in the sentence "The model has two main parameters: N, the number of components in a system, and K, the degree of interaction between components." the letter "K" similarly to "N" should be in italics

Page 8, Figure 2 caption

Brackets in the text of manuscript should be standardized, namely parentheses () used for additional information in parentheses and square brackets [] for cited literature. Please check the entire manuscript in this context.

Page 19

To the best of my knowledge should be ,,References” instead of ,,Reference”

Reviewer 2 Report

Dear Authors, 

I have some suggestions to improve their work.

The manuscript is interesting and well-designed; some significant points should be addressed before the manuscript can be considered for publication.

- Throughout the text, check the conjugation of the verbs.

- The paper needs to be checked carefully for typos and grammatical errors.

- Keep the same font style and size throughout the manuscript.

For example, in the figure caption of figure 3, there are some errors like the above mentioned, so the corrected text is Figure 3. Evolutionary walks on NK landscapes. (A) We simulated the adaptive evolution of a microbial population on ten NK landscapes (Fig 2) with the length of binary strings N=10 and the number of interacting sites ranging from K=0 to K=9 (x-axis). Populations (consisting of n=100 binary strings) evolved according to three evolutionary algorithms (Fig 1): fitness-seeking (top panel), novelty-seeking (middle panel), and hybrid walks (bottom panel). Strings mutated at an average rate of one-bit flip per haplotype per generation. Evolutionary walks were repeated 50 times for each algorithm and at each level of landscape complexity (K). Each point represents the number of generations (y-axis) either when the population reached the global fitness peak (g<100) or when the simulation reached the generation limit (g=100). Fitness-seeking walks reached global fitness peaks with increasing difficulty as the landscape became more complex (top panel). Novelty-seeking walks reached fitness peaks by chance and were not affected by landscape complexity (middle panel). Hybrid walks, including fitness- and novelty-seeking walks (chosen at random with probability p=0.5), could reach fitness peaks with high success rates even on the most complex landscapes (bottom panel). (B) Fitness values (y-axis) of the fittest individuals at each generation (x-axis) were plotted for populations evolving under three search algorithms (panels) at four complexity levels (K=0, 2, 5, and 10). Populations evolving with fitness-seeking walks increased fitness monotonically but tended to reach only local fitness peaks, especially when landscapes were complex (top panel). Populations evolving with novelty-seeking walks fluctuated in fitness values without consistently reaching global or local peaks (middle panel). Populations evolving with hybrid fitness- and novel-seeking walks consistently avoided the local fitness peaks and reached the global fitness peaks (bottom panel). Note that the novelty of a binary string was defined as the unique position of the string in an N-dimension hypercube (see Algorithms & Methods).

The manuscript submitted needs to be entirely in the format suggested by MDPI.

In the References section, there is no number for each article. 

In the text, the citation of the articles does not correspond to what was requested by the journal. 

The references are not in the format and order suggested.

Abstract

-The abstract should be a total of about 200 words maximum.

Introduction

This section provides a comprehensive overview of host-pathogen coevolution. Moreover, parts of the introduction already describe results and would better fit the results or discussion section. I would recommend providing only a summary of the findings at the end of the introduction and leaving the rest for discussion.

Results

The manuscript is challenging to read, and I propose rewriting the figure caption because the information is repetitive between the results text and the figure caption. In the figure captions, the main emphasis should be on relevant data.

Round 2

Reviewer 2 Report

Dear authors,

All suggestions were adequately addressed, thank you.

Please wait for the editor's comments and the publisher's indications for the acceptance and publication of the article.